# Monocellular and Multicellular Parasites Infesting Humans: A Review of Calcium Ion Mechanisms

**DOI:** 10.3390/biomedicines14010002

**Published:** 2025-12-19

**Authors:** John A. D’Elia, Larry A. Weinrauch

**Affiliations:** E P Joslin Research Laboratory, Beth Israel Deaconess Kidney and Hypertension Section at Joslin Diabetes Center, Department of Medicine, Harvard Medical School, Boston, MA 02215, USA

**Keywords:** calcium ion channel blocker, calcium ions, parasites, calcium release operated channels, voltage gated channels, calcineurin inhibition

## Abstract

Calcium (Ca^2+^) is a signal messenger for ion flow in and out of microbial, parasitic, and host defense cells. Manipulation of calcium ion signaling with ion blockers and calcineurin inhibitors may improve host defense while decreasing microbial/parasitic resistance to therapy. Ca^2+^ release from intracellular storage sites controls many host defense functions (cell integrity, movement, and growth). The transformation of phospholipids in the erythrocyte membrane is associated with changes in deformability. This type of lipid bilayer defense mechanism helps to prevent attack by *Plasmodium*. Patients with sickle cell disease (SS hemoglobin) do not have this protection and are extremely vulnerable to massive hemolysis from parasitic infestation. Patients with thalassemia major also lack parasite protection. Alteration of Ca^2+^ ion channels responsive to environmental stimuli (transient receptor potential) results in erythrocyte protection from *Plasmodium*. Similarly, calcineurin inhibitors (cyclosporine) reduce heart and brain inflammation injury with Trypanosoma and Taenia. Ca^2+^ channel blockers interfere with malarial life cycles. Several species of parasites are known to invade hepatocytes: Plasmodium, Echinococcus, Schistosoma, Taenia, and Toxoplasma. Ligand-specific membrane channel constituents (inositol triphosphate and sphingosine phospholipid) constitute membrane surface signal messengers. Plasmodium requires Ca^2+^ for energy to grow and to occupy red blood cells. A cascade of signals proceeds from Ca^2+^ to two proteins: calmodulin and calcineurin. Inhibitors of calmodulin were found to blunt the population growth of Plasmodium. An inhibitor of calcineurin (cyclosporine) was found to retard population growth of both Plasmodium and Schistosoma. Calcineurin also controls sensitivity and resistance to antibiotics. After exposure to cyclosporine, the liver directs Ca^2+^ ions into storage sites in the endoplasmic reticulum and mitochondria. Storage of large amounts of Ca^2+^ would be useful if pathogens began to occupy both red blood cells and liver cells. We present scientific evidence supporting the benefits of calcium channel blockers and calcineurin inhibitors to potentiate current antiparasitic therapies.

## 1. Introduction

Climate change, travel, migration, and the development of multiple drug resistance have exposed new populations to infectious disease. This creates great social and economic pressure to develop more effective (and expensive) antibiotics. One area that has not been explored, however, is a search for adjuvant or adjunctive therapies that might improve the effectiveness of medications already in use.

Antimicrobial usage increases within each geographic region based on exposure and availability. This has been true of antiparasitic, antibacterial, and antiviral medications. Antimicrobial resistance is a complication of inadequate exposure to medications in large populations. There are multiple reasons for the inadequate treatment of infectious agents. Among these are costs (price, side effects, and availability) and inadequate exposure (dosage and duration). For some infections/infestations, we may overcome medication resistance with adjuvant or adjunctive therapies. An adjuvant is a substance, or a combination of substances, used to increase the efficacy or potency of another drug. Adjuvants have an insufficient benefit when used alone for the problem under study. Adjuvant or adjunctive therapy generally assists the primary therapy. Such therapies, identified for more than a century (e.g., acidification of urine to potentiate treatment of urinary infections; use of calcium channel blockade to increase circulating cyclosporine levels; neprilysin inhibition to potentiate angiotensin receptor blockade; or ice when added to nonsteroidals and rest). Clinicians rarely use these terms except in oncology or psychiatry.

Three widespread infectious diseases (tuberculosis, malaria, and schistosomiasis) have developed resistance to effective antibiotics [1,2,3]. The means by which pathogens become resistant to antibiotics may involve Ca^2+^ ion channels. Pathogen mutation complicates efforts to control disease spread related to climate change, international travel, and malnutrition in centers of increasing population density [4,5]. Populations may be attacked by multiple strains [6,7] or by simultaneous occurrence of viruses that increase pathogenicity of human parasites (HIV and visceral leishmaniasis [8]; COVID-19 and malaria [9,10,11]). The additional threat of warming climates has allowed vectors of parasitic disease to move from the tropics to more temperate locations [12]. Anecdotal reports of the Anopheles mosquito in southern locations of the United States of America will eventually result in the reappearance of clinical malaria after an infestation-free period of 70 years. Avian malaria has been described as responsible for escalating bird mortality in the Hawaiian Islands (Kauai) [13].

Recent research has focused on the role of calcium ion movements associated with parasitic infestation and antiparasitic drug resistance. In this review, we focus on the use of adjunct or adjuvant therapy to improve host defense once invaded by human parasites. We review biological-calcium-ion-related mechanisms used by pathogens to persist in human hosts, as well as mechanisms used by human hosts to survive infestation.

Perhaps the most infestation-susceptible, non-neoplastic state is that of adult cystic fibrosis, in which mutation of a transmembrane protein channel no longer allows for the essential movement of chloride (Cl^−^) and water. The result is a thick, dry mucus blocking tubules in the lung’s branchial epithelium, resulting in bronchiectasis. In addition, exocrine structures of the pancreas are injured by chronic inflammation, which eventually leads to damage to Beta cells in the Islets of Langerhans. The chronic inflammation reaction eventually causes insulin-dependent Type 1 diabetes mellitus.

With certain parasite attacks, calcium levels in host plasma, tissue fluid, or intracellular spaces may increase as a defensive measure. Susceptibility to infection is associated with the movement of Ca^2+^ from intracellular storage sites into plasma, tissue fluids, and bronchial epithelial cells [13]. Cellular concentration of Ca^2+^ in the injured bronchial epithelium is elevated beyond that of the normal tissue [14]. However, the defense mechanism by which the patient (host) survives involves a calcium-signaled interleukin-dependent inflammatory cascade (phagocytic neutrophils, macrophages, and cytochrome-bearing lymphocytes) [15]. There may be a role for calcium channel blockers in protecting the cystic fibrosis patient from damage due to chronic excretory duct obstruction resulting from “over-activated” store-operated calcium intake. When the critical protein structure responsible (orai1) for over-activated calcium intake has been treated with an inhibitor, the chronic pancreatic tissue injury pattern will stop [15,16]. This experimental model may indicate a role for future calcium channel blocker studies.

The concentration of Ca^2+^ in plasma is measured in the millimolar range, while it is in the nanomolar range in the cytoplasm of cells with nuclei (eukaryotes). Parasites require a narrow range of concentrations of calcium in their cytoplasm. If the level rises, the parasite is at risk for apoptosis. To move calcium into the circulation of the host, pumps are needed since the plasma level is higher than the cytoplasm level. Calcium is an essential agent in cellular homeostasis, with signals that promote downstream reactions in Trypanosomes [17,18].

Since the cell growth of Toxoplasma may be controlled by both external and internal Ca^2+^ channels, this factor may be a focus for calcium-channel-blocking therapy. It has been suggested that blocking calcium signaling through transient receptor potential channels (TRP) might be a useful target to decrease pathogenicity [19,20]. In addition to calcium-channel-blocking medications, calcineurin inhibitors may play a role in benefiting the host in a different manner. One such way is to move calcium into hepatic cellular storage sites (the endoplasmic reticulum and mitochondria). The availability of larger calcium stores may permit deployment to counter parasitic liver attack. With the aid of mitochondria that buffer the intensity of calcium flow, a controlled release of Ca^2+^ from intracellular stores may be effectively used for defense without fear of local tissue damage [21]. This balanced response has been documented in protection from filarial parasites, resulting in their elimination [22].

Parasites known to attack the liver include Echinococcus, Nematodes, Plasmodium, and Schistosomes. Of the four, Nematodes and Plasmodium may also attack the central nervous system. In Taenia solium infestation of the central nervous system, there are several asymptomatic years while the quiescent cestode worm resides within a formed cyst. If the parasite’s life cycle is suddenly terminated by treatment or cell defenses, then an allergic reaction may occur as breakdown products trigger inflammatory cytokines and healing [23]. Healing mechanisms may require movement of calcium from exterior and interior sites with a process involving toll-like receptors in association with Ca^2+^ signaling, made possible by movement through external (SOCE) and internal (endoplasmic reticular and mitochondrial) sources [24,25,26]. Tight control of inflammation cytokines is needed because damage to healthy tissue can occur if a recovery process is too robust, which is associated with a level of cytosolic Ca^2+^ that is too high for cell-surface membrane structures.

## 2. Cell Regulation Pathways Associated with Movement of Calcium

Populations naïve to parasitic infestation may suffer serious consequences with massive hemolysis and increased mortality in the case of malaria. Hemoglobin and free myoglobin are toxic to kidney tubules [27,28]. Acute renal tubular necrosis following hemolysis or rhabdomyolysis may have secondary injury from calcium phosphate crystal deposition (nephrocalcinosis) in addition to the primary cause of acute tubular necrosis [29]. Healthy younger patients usually recover from short-term malaria-induced acute tubular necrosis, but underlying malnutrition and unsanitary public services may be associated with prolonged recovery [30].

Ca^2+^ release from intracellular storage sites controls many host defense functions (integrity, movement, and growth). The transformation of phospholipid [31] in the erythrocyte membrane is associated with changes in deformability. This type of lipid bilayer defense mechanism helps to prevent attack by *P. falciparum*. Patients with sickle cell disease (SS hemoglobin) lack such protection and, therefore, are extremely vulnerable to massive hemolysis from parasitic infestation. Patients with thalassemia major also lack parasite protection. Gene mutations associated with greater protection from parasite injury are documented in the sickle cell trait by expression of SA hemoglobin and in thalassemia minor by expression of an alteration in a single globin chain [32].

## 3. Regulation of Intracellular Ca^2+^ May Require Channels Activated by Ca^2+^ Utilizing ATPase Enzyme [33]

Intracellular regulation of Ca^2+^ concentration is essential for cellular movement in the single or multicell parasite, as well as the human organism, with respect to the cells, organs, and whole body. Ca^2+^ movement out of the endoplasmic reticulum is noted in the contraction/relaxation of skeletal and cardiac muscle. For muscle function, energy is supplied by the action of ATPase on ATP for the release of high-energy phosphate. Specialized membrane constituents in Ca^2+^-regulated movement may contain phospholipid derivatives (sphingosine). Excessive Ca^2+^ movement is buffered by mitochondria [33,34,35,36]. Two systems of Ca^2+^ movement consist of one operating at normal-concentration oscillation and one operating at a high concentration with frequent concentration oscillation. Loss of efficient contraction/relaxation of both skeletal and cardiac muscles is a direct consequence of the delayed movement of Ca^2+^ out of and into the cell stores. [37,38,39,40]. The locomotion of certain parasites swimming in fresh water requires Ca^2+^ oscillations. High Ca^2+^ levels may result in tetany; low Ca^2+^ levels are associated with flaccidity.

Ca^2+^ release from intracellular storage controls cell growth, movement, and death via apoptosis [39]. Intracellular Ca^2+^ levels change when life cycle stages require a “swimming” movement within a liquid medium. Schistosome cercariae, after release from their snail vector, must reach an arm or a leg of a human host in fresh water [40]. The contraction/relaxation of swimming muscles will typically be measured by the ability of intracellular Ca^2+^ to oscillate continuously from the endoplasmic reticulum to the cytosol and back again. However, if c Ca^2+^ levels do not oscillate, then elevated concentrations are associated with tetany, and reduced concentrations are associated with flaccidity, i.e., paralysis. Spastic paralysis of cercariae has been demonstrated under experimental conditions. Ca^2+^ signaling in malaria parasites demonstrates a concentration peak in red blood cells just prior to their membrane rupture [41,42]. Red blood cell membrane rupture releases schizonts into the circulation. Schizonts mature into trophozoites, seen microscopically in the ring stage.

## 4. Cell Coordination Mechanisms Employ Sodium (Na^+^)/Calcium (Ca^2+^) Exchange Signals

Parasite adjustments to the human life cycle may result from mutations occurring over short or long timeframes. Short-term adjustment includes mosquito blood meals shortly after human host meals. Melatonin is a hormone whose synthesis and secretion are associated with the timing of hours of sleep. Humans use a pharmaceutical form of this hormone to resolve insomnia. Biosynthesis and secretion of melatonin are associated with the release of Ca^2+^ from cell storage sites [43]. Other functions that may follow the release of Ca^2+^ into the cytosol may involve inositol-3 phosphate as a membrane-surface signal associated with multiple functions in multiple locations. Enzyme reactions that may be initiated by mechanisms involving inositol-3 phosphate/Ca^2+^ include cell growth; synthesis of ATP; repair of DNA; and cell maturation into specific functions, such as insulin signaling [44,45]. If these interactions are intended for host survival, then parasites and vectors must adjust to the rhythm of the hour of the day. The host’s timing of meals or periods of rest sets the rhythm. That Anopheles mosquitoes coordinate blood meals after hosts have taken their meals is a natural consequence. Long-term mutations may be more affected by a slowly changing climate. Those inositol phosphate enzyme systems that preserve the capacity for reproduction enable the synthesis of ATP or repair nuclear DNA [44,45]. The fact that intracellular Ca^2+^-containing stores in Leishmania donovani may be available for the regulation of a wide-ranging community of enzyme systems has been the subject of investigation [46,47,48]. Subsequently, identification of intracellular Ca^2+^ release channels has also been described in Schistosome mansoni [49]. In mammals, intracellular Ca^2+^ release channels include ryanodine receptors, two-pore Ca^2+^ (TPC) channels, intracellular transient receptor potential (Trp) channels, two Ca^2+^ influx channels, and voltage-gated and plasma membrane Trp channels, as well as frequently reported inositol triphosphate receptors [44,45]. Ryanodine receptor channels are best known for their role in Ca^2+^ release from the endoplasmic reticulum during excitation/contraction coupling of cardiac and skeletal muscle [50]. Hydrolysis of ATP by ATPase is a critical step in the release of high-energy phosphate [34,40]. This source of energy needed to move Ca^2+^ against a concentration gradient is critical in muscle function.

*Plasmodium falciparum* can simultaneously intake nutrients and invade the erythrocyte membrane bilayer]. A family of proteins (rhoptry) cooperates in this synergy [51,52,53]. Defense mechanisms, which have evolved over long periods of time, are associated with the movement of Ca^2+^. These defense maneuvers would be of little value if Ca^2+^ concentrations rose high enough to injure the host while eradicating the parasite. Coordination of responses, along with a careful handling of risks, may provide protection for the host, along with a controlled reduction in the parasite population. Coordination of survival maneuvers for *Plasmodium falciparum* involves a surface anion channel that allows for the elimination of waste along with the intake of nutrients and ions, but not osmotically active Na^+^ [51,53]. Uncontrolled intake of Na^+^ requires a compensatory intake of water. The resultant swelling is a risk for injury to both the parasite and red blood cells. Sodium/glucose cotransporters employ a synergy of two independent functions for the intake (or reabsorption) of a nutrient fuel, while Na^+^ moves in the opposite direction [54]. In addition to cell wall strength, a certain amount of flexibility is achieved through a process that involves Ca^2+^ binding [55].

Multiple aspects of the cardiac Na^+^/Ca^2+^ cotransporter have been thoroughly reviewed by Xue and colleagues [56]. Cardio-myocyte contraction/relaxation requires instantaneous Ca^2+^ signaling. A practical relationship promoting this vital function is the proximity of the plasma membrane to the endoplasmic reticulum of the myocyte. Given the rapidity of contraction/relaxation, coordination of the exchanger is required, as well as the routine movement of standard concentrations of intracellular Ca^2+^ caused by a plasma membrane Ca^2+^ ATPase [34,40]. Loss of efficient movement in and out of the endoplasmic reticulum has been demonstrated in experimental animals with cardiomyopathy associated with diabetes mellitus [37,38,39].

The most important metabolic susceptibility for parasite infestation is diabetes mellitus. Other interesting features of the cardiac Na^+^/Ca^2+^ exchanger are its presence in the retina of the eye [57], as well as the distal tubule of the kidney [58]. Since these locations are of concern in terms of avoiding several complications of diabetes mellitus, it is important to test function in an experimental model of insulin-deficient type 1 diabetes mellitus [59]. Compared to controls, streptozotocin diabetic rats after aortic obstruction demonstrated decreased heart rate, decreased peak left ventricular pressure, increased diastolic pressure, and prolonged left ventricular relaxation time. These abnormalities of the experimental db/db diabetic rats were corrected following treatment with insulin. Other studies of streptozotocin diabetic rats compared to controls demonstrated voltage losses of 50% for Na^+^/Ca^2+^ exchangers, as well as a 30% loss of protein + mRNA for these exchangers. All deficiencies in these experimental db/db diabetic rats were corrected following treatment with insulin. The possibility of complications associated with chronic hyperglycemia raises the question of parasite attack on the pancreas of the host. Of the four parasites that commonly occupy the liver of the human host, three are associated with hyperglycemia. *Echinococcus granulosis*, *Taenia solium*, and *Toxoplasma gondii*, but not *Schistosome mansoni*, have a 6–11% risk of hyperglycemia [60,61,62,63].

Other points of interest in Na^+^/Ca^2+^ exchangers inside the heart or outside the cardiac space are useful in understanding the regulation/coordination of cell metabolism [64,65,66,67,68,69]. Ca^2+^-related cell functions include the generation of ATP, the repair of DNA, and the initiation of both inflammation and immunity responses.

## 5. Cell Immunity Organized by Calcium-Release-Activated Calcium Channels and Purinergic Signals

Stimuli that may initiate the construction and/or use of an ion channel may be an early step toward the expression of an immune-reactive system. Temperature, pH, pain, and mechanical pressure (stretch) are known as nociceptive defense signals. For example, an anion channel in red blood cells is activated by stretching, whereas the unstimulated red blood cell has no anion channel in its lipid bilayer [70]. Following the invasion of host red blood cells by *Plasmodium falciparum*, two anion channels emerge in infected erythrocytes [71]. Of the two channels, the voltage-gated one demonstrates activation by the presence of the parasite. The second anion channel, which is not voltage-gated, is not directed by the presence of the parasite. This suggests that the second channel is involved in housekeeping chores like elimination of waste, along with the intake of nutrients, while the first is upregulated by the parasite for the purposes of survival through specific defense.

The interaction of increased Ca^2+^ concentration with immune/inflammation factors has been described in association with Ca^2+^ movement from the endoplasmic reticulum, as well as from the exterior via store-operated calcium entry (SOCE) channels [24,72,73,74]. T lymphocyte function in some instances is closely related to Ca^2+^ intake by calcium-release-activated calcium (CRAC) channels [75,76]. Release of intracellular ATP activates T lymphocyte migration toward antigen-bearing cells. The ATP energy source is from the mitochondria [21,34,40]. Associations with this Ca^2+^-dependent channel include the immediate release of toxic granules by CD8 T cells. Unlike the functions of many T lymphocyte groups, the functions of CD8 T cells are not suppressed by suppression of Ca^2+^ intake at the CRAC channel. Purinergic signaling, which involves adenosine, ATP, and ADP, is associated with the regulation of immune/inflammation reactions [73,76,77,78,79,80]. Defense systems, which may utilize an increased concentration of Ca^2+^ as their first responder, must also be capable of limiting collateral damage to healthy tissue.

Purinergic signals have the dual responsibility of initiating inflammation for protection from invasive viruses, bacteria, fungi, or parasite species. In addition to directing inflammation for lethal damage to pathogens, there is also the responsibility of protecting host cells from collateral damage from systems employed by the defense. Studies of several parasite species have identified interference with purinergic signaling at the center of aggressive pathogen infestation [81,82,83,84,85,86,87,88,89]. After the introduction of Leishmania through the skin by the bite of a sand-fly, the method of attack on the epidermal and gastrointestinal systems of the human host involves disabling the purinergic signaling of the host at the level of enzymes that hydrolyze ATP [77]. Ectonucleotidase enzymes release adenosine from ADP and ATP [78]. Free adenosine enhances the efficacy of pathogen invasion by inhibiting host-defense-related generation of neutrophil phagocytes and regulatory T cells [73,79]. ATP and adenosine are eventually recognized as being in opposition, i.e., positive function vs. inhibition of host purinergic signals [76].

Purinergic signals are used extensively in the Leishmania community. *Leishmania donovani*, the most frequently encountered member, causes skin lesions. *Leishmania amazonensis*, the most aggressive member, causes lesions of the gastrointestinal tract. *Leishmania amazonensis* has been found to generate greater amounts of adenosine by more rapid hydrolysis of ATP than Leishmania donovani [80,81]. When *Toxoplasma gondii* is found in immunocompromised patients, there is concern for activation of dormant cysts in brain tissue, leading to fatal encephalitis. In this instance, the original infestation may have occurred when the host was fully immunocompetent. Wistar rats injected with Toxoplasma gondii demonstrated ectonucleotide enzymes in circulating lymphocytes, as well as products of ATP hydrolysis in brain samples [82,83]. *Trypanosoma cruzi* infestation of the heart causes a potentially fatal cardiomyopathy known as Chagas disease. Levels of ectonucleotidase enzymes reflect the activity of the parasite [76]. There is a relatively dormant indeterminate stage during which levels of ATP are relatively high, and levels of adenosine are relatively low because the activity of ectonucleotidase enzymes is low [76]. In addition, elevated levels of ectonucleotidase enzymes correlate with the severity of myocarditis in individuals with long-standing Chagas disease [84]. The same observation applies to individuals with malaria, whose disease activity is associated with increased activity of enzymes that can hydrolyze ATP, thereby releasing adenosine [85]. There is evidence that the morbidity of two worm infestations of the liver operates through altered purinergic signaling. The activities of *Schistosome mansoni* and *Fasciola hepatica* have been quantified in connection with the hydrolysis of ATP, which means interference with purinergic signaling by adenosine [86,87]. Studies of the Plasmodium family of malaria-infestation agents have reported purinergic signaling to be essential for attack on red blood cells [76]. *Plasmodium falciparum* has been shown to have limited energy for entry into red blood cells with limited powers of reproduction when exposed to an inhibitor of purinergic signaling [84,88,89].

Immune complications resulting in anaphylaxis have been reported in patients with long-standing infestation with *Echinococcus granulosis*. These individuals may have had asymptomatic liver cysts (hydatid cysts) for many years. Calcified cysts do not generate antibodies or lymphocyte responses. However, a cyst may rupture, sending foreign proteins into the biliary drainage system, the gastrointestinal tract, pulmonary tissue, peritoneal cavity, or general circulation [90].

## 6. Cell Resistance to Parasite Infestation Enhanced by Antibiotics Through Ca^2+^ and Parasite Cell Resistance to Antibiotics Enhanced by Calcium Channels

Plasmodium falciparum requires Ca^2+^ for energy to grow and to occupy red blood cells. A cascade of signals proceeds from Ca^2+^ to two proteins: calmodulin and calcineurin. Inhibitors of calmodulin were found to blunt the population growth of *Plasmodium falciparum* [91]. An inhibitor of calcineurin (cyclosporine) was found to retard the population growth of both *Plasmodium falciparum* and *Schistosoma mansoni* [91,92]. On the other hand, cyclosporine appeared to advance the virulence of *Trypanosome cruzi*. *Leishmania donovani* responded to cyclosporine in both positive and negative ways. Calcineurin also has control of sensitivity to antibiotics, as well as resistance to antibiotics [93]. After exposure to cyclosporine, a calcineurin inhibitor, the liver directs Ca^2+^ ions into storage sites in the endoplasmic reticulum and mitochondria [20]. Storage of large amounts of c Ca^2+^ is useful as ammunition if pathogens begin to occupy red blood cells or liver cells [94]. Four species of parasites are known to invade hepatocytes: *Echinococcus granulosis*, *Schistosoma mansoni*, *Taenia solium,* and *Toxoplasma gondii*.

Regarding the store-operated Ca^2+^ entry in *Trypanosoma equiperdum*, physiological evidence of its presence can be found in Table 1, which summarizes targets for treatment of *Plasmodium falciparum* malaria based on its life cycle in the blood or liver. Of note is the acknowledged benefit provided by the addition of the calcineurin inhibitor cyclosporine to standard malaria treatment antibiotics, such as atovaquone/proguanil for the hepatocyte phase or artemether/lumefantrine for the erythrocyte phase. Table 2 lists mechanisms utilized by antibiotics and calcineurin inhibitors in host defense from malaria caused by *Plasmodium falciparum*.

Prior studies of *Leishmania donovani* had found intracellular stores of Ca^2+^ to be involved with the function of enzymes [48]. Ca^2+^ channels are the route used for the efflux of prescribed medications by antibiotic-resistant pathogens like *Mycobacterium tuberculosis* [95,96]. When employed by *Plasmodium falciparum*, the intracellular concentration of chloroquine is lowered before the intended lethal effect [97]. Since this antibiotic efflux occurs through a Ca^2+^ channel, verapamil may be effective in reversing resistance to chloroquine [98,99]. We found no human clinical trials of CCBs for patients with malaria listed in clinicaltrials.gov. Given the relative safety at a low cost, a clinical trial of CCBs (amlodipine, diltiazem, nifedipine, and verapamil) as adjuncts to standard anti-malarial drugs might be worthwhile. Since CCBs are now available as generic medications, no commercial pharmacological sources of funding can be expected. Immunologic and genetic pathways may direct future malaria research in the immunocompromised host [100,101,102]

Certain intestinal roundworms are the target of albendazole and mebendazole, which use the same two mechanisms of action. These two drugs counterattack both the larval and adult life cycle stages of Nematodes (*Ascaris lumbricoides*), hookworms (*Necator americanus* and *Ancylostoma duodenale*), and pinworms (*Enterobius vermicularis*). The major mechanism of action is the disruption of the absorption of glucose fuel through tubules, leading to the loss of mitochondrial production of high-energy phosphate from ATP [103,104]. The minor mechanism of action is interference with parasite movement caused by the interruption of the Ca^2+^/acetylcholine relationship. Greater/lesser brain or peripheral nervous system function requires increased/decreased activity of cholinesterase associated with coordinated oscillations of Ca^2+^ concentrations [105].

## 7. Cell Movement of Calcium from One Compartment to Another May Involve a Na^+^/H^+^ Exchanger

Totally direct Na^+^/Ca^2+^ exchangers are in use in *Plasmodium falciparum* [17]. Sodium/calcium and sodium/hydrogen exchangers may work in tandem with Ca^2+^ movement. This more complicated arrangement for intracellular movement of Ca^2+^ has been described for *Taenia coli*, *Trypanosoma brucei*, *Leishmania donovani*, and *Toxoplasma gondii* [106,107,108,109,110,111]. In this setting, the first step is a sodium/hydrogen exchange. After subsequent adjustments, including pH, the movement of Ca^2+^ may take place. An increase in membrane permeability may be required. Neutralization of the cytosol of parasites bearing acidisomes may be the focus of anti-Trypanosome and anti-leishmanial medications [109,110].

Table 1 Summarizes targets for treatment of *Plasmodium falciparum* malaria based on the life cycle in blood or liver. Of note is the acknowledged benefit provided by the addition of the calcineurin inhibitor, cyclosporine, to standard malaria treatment antibiotics such as atovaquone/proguanil for the hepatocyte phase or artemether/lumefantrine for the erythrocyte phase.

Table 2 lists mechanisms utilized by antibiotics and calcineurin inhibitors in host defense from malaria caused by *Plasmodium falciparum*.

Table 3 identifies organ systems attacked by several parasites. There are four species listed as Liver since their occupation may have pathological consequences (Echinococcus, Schistosome, Taenia, and Toxoplasma). There are six species listed as Other Viscera (Cryptosporidium, Echinococcus, Leishmania, Nematode, Schistosome, and Trypanosome). The most life-threatening species have been discovered attacking structures in the brain (Nematode, Plasmodium, Toxoplasma, and Trypanosoma). Since parasites have nothing to be gained by the death of their host, we might question whether the hosts were immunocompromised by malnutrition.

Table 4 summarizes ion channel mechanisms in several connections used to study mechanisms of human hosts in parasite infestation, as well as parasite defense from lethal counterattacks.

## 8. Possible Future Developments

Multiple roles of c Ca^2+^ channel signaling in human host/parasite interaction offer possibilities for efficient control of infestation by parasites. Sufficient human and animal data supports roles for standard c Ca^2+^ channel blockers (verapamil, amlodipine, and diltiazem) as adjuncts to standard therapy. In addition, Ca^2+^ channel blocking in an experimental animal study has been found with cholesterol-lowering statin (atorvastatin, simvastatin, and rosuvastatin) medications. A longer-lived liver-stage life cycle *Plasmodium falciparum* vaccine of high-grade efficiency has reached the human study level [112]. Further success in vaccine efforts may relieve constant high pressure to develop anti-malarial drugs. The USA Food and Drug Administration has recently released an oral agent, Nitisinone, which, in small doses, is a more effective insecticide than ivermectin [112]. Nitisinone attacks the mosquito digestive enzyme 4-hydroxyphenyl pyruvate dioxygenase, which is useful in tyrosine detoxification.

Ca^2+^ channels, which involve intake from outside of the cell, involve a filter for the exclusion of toxins, a gate to permit ion flow, and a pore to control the destination. Voltage-gated Ca^2+^ channels (VGCCs) closely monitor electrochemical potential in association with Ca^2+^ concentration gradients. For cell entry down the steep gradient, VGCCs are highly efficient. For Ca^2+^ ion exit against a thousand-fold gradient, a strong pump, utilizing a great deal of energy, is required. The main pump is the Na^+^/Ca^2+^ exchange pump [113]. Another pump is the plasma membrane c Ca^2+^ ATPase, a secondary contributor to low-Ca^2+^-level homeostasis [114].

A unique pump, containing the phospholipid sphingosine and preserving the low concentration level needed for Ca^2+^ to function as a second messenger, has been reported in a Trypanosome-based study [115]. Trypanosomes and Leishmania contain L-type VGCCs. An effective antiparasitic medication in this connection is miltefosine, the structure of which is much like that of sphingosine [38,116,117]. This is also the domain of the attachment of dihydropyridine CCBs (amlodipine, felodipine, and nifedipine), which is far removed from the domain of the non-dihydropyridine CCB verapamil [117]. The ability to image reaction sites through developments in electron microscopy has allowed research groups to pinpoint reaction sites with the point of attachment of CCBs [118,119].

Table 5 lists Ca^2+^ channel mechanisms. High-voltage L-type Ca^2+^ channels are found in Leishmania, Schistosomes, and Trypanosomes [120,121,122,123,124]. A unique mechanism for Ca^2+^ entry through the plasma membrane involves a transient receptor potential (TRP) channel, which has been reported for Toxoplasma [120]. A TRP activated by praziquantel used in the treatment of a schistosome has also been reported [122]. Ligand-related Ca^2+^ channels include sphingosine for Leishmania and inositol triphosphate for Trypanosomes [115,124]. Table 6 lists some ion transit mechanisms that have been reported in the human parasite literature.

Approximately 400 genes have been identified to control chloride channels within cells. Destruction of the cell envelope has been associated with internal chloride channel pathology, which increases chloride concentration. Pharmacotherapeutic and biomimetic technologies have been employed to reproduce this process in treating gram-negative bacterial or cancer cell invasion [125,126]. Biomimetic ion channel technology may be anticipated to be widely evaluated for antiparasitic therapy aiming to induce apoptosis, altered autophagy, and/or dysfunction of lysosomes and mitochondria [127]. To our knowledge, no biomimetic technology studies have yet been reported for the treatment of human parasites.

This review focuses on known ion channels controlling Ca^2+^ fluxes and their relationships to parasite and human physiology. Newer biological tools will enable the development of probes to explore these relationships further. It is our hope that such exploration will expand our knowledge of targets for adjunctive or curative biologic therapy.

## Figures and Tables

**Table 1 biomedicines-14-00002-t001:** Predominant organ systems clinically affected by parasites that commonly infect humans.

	CNS	Heart	Liver	Viscera (Other)	Blood	Skin
*Plasmodium falciparum*	+		++		++	
*Leishmania*				++		++
*Trypanosoma brucei, cruzi,*	++	++				
*Toxoplasma gondii*	++					
*Schistosoma haematobium*				++ (bladder)		
*Schistosoma japonicum*			++	++		
*Schistosoma mansoni*			++			
*Taenia solium (cestode)*	++		++	++		
*Cryptosporidium*				++ (lung, GI)		
*Echinococcus*			++	++		

+ indicates potential serious infection complication.

**Table 2 biomedicines-14-00002-t002:** Potential therapeutic interference with *Plasmodium falciparum* cycles.

Initial Lymphohematogenous Phase: Injection of sporozoites by Anopheles mosquito	Potential interference By environmental control By vaccine
Hepatic Phase: Maturation of Plasmodium sporozoites into merozoites and formation of clusters called schizonts	Quinine and atovaquone/proguanil inhibit parasite use of dihydrofolate reductase, DNA synthesis, and use of oxygen caused by hemoglobin binding Calcium required for release of merozoites from schizonts—intake blocked by cyclosporin (calcineurin inhibition)
Hematogenous Phase: Rupture of schizonts permits merozoites to enter bloodstream and parasitize erythrocytes	Trophozoite damaged by oxygen radicals: quinine and artemether/lumefantrine Trophozoite damaged by inhibition of synthesis of protein and nucleic acid by chloroquine Trophozoite extrusion of antibiotics blocked by verapamil (calcium ion channel blockade) Cyclosporines are potent inhibitors of intraerythrocytic parasite growth
Maturation of parasites (merozoites to trophozoites, schizonts) with subsequent rupture of erythrocytes	
Release of merozoites into the bloodstream: Acute malaria hemolytic crisis	

**Table 3 biomedicines-14-00002-t003:** Treatment mechanisms in *Plasmodium Falciparum* malaria.

Human Host Anatomy	Life Cycle Phase	Therapeutic Agent Mechanism
Hepatocyte/erythrocyte	Merozoites and schizonts Merozoites in red blood cells	Cyclosporine/chloroquine Atovaquone/proguanil Impairs calcium peak
Erythrocyte	Trophozoites	Chloroquine inhibition of protein and nucleic acid synthesis Artemether/lumefantrine binds to heme, generating oxygen radicals
Anopheles mosquito	Life cycle phase	Therapeutic agent mechanism
Salivary gland	Sporozoites	Vaccine (trial)

**Table 4 biomedicines-14-00002-t004:** Calcium-related mechanisms in parasite infestations of human hosts.

**A.** **Defense** CFTMP (cystic fibrosis transmembrane protein) chloride channel Mitochondria as calcium buffer ORAI-1 (over-activated intake) of calcium TRP (transient receptor potential) channel Stretch-activated (PIEZO) **B.** **Regulation** ATPase (adenosine triphosphate) hydrolysis enzyme ER (endoplasmic reticulum) calcium storage Mitochondria calcium storage **C.** **Coordination** MSS (membrane surface signals) Inositol triphosphate (ITP3) Sphingosine (derivative of phospholipid) Na^+^/Ca^2+^ transporter Congestive heart failure **D.** **Immunity** SOCE (store operated calcium entry) channel Purinergic Signals ATP ADP Adenosine **E.** **Antibiotic Resistance** Calcineurin Inhibitors (cyclosporine) Adjunct to anti-malarial drugs Calcium channel blockers (amlodipine, verapamil) Adjunct to anti-malarial drugs **F.** **Ion Exchangers in Parasites** Totally Direct Action Na^+^/Ca^2+^ exchangers *a.* *Plasmodium falciparum* *b.* *Taenia coli* Intermediate Direct Action Na^+^/H^+^/Ca^2+^ Exchanger *a.* *Trypanosome brucei* *b.* *Leishmania donovani* *c.* *Toxoplasma gondii* Totally Indirect Na^+^/acetylcholine exchanger *a.* *Ascaris lumbricoides*

**Table 5 biomedicines-14-00002-t005:** Targeting calcium ion channels of parasites: potential to address the capacities to survive, invade, move, multiply, and secrete proteins.

**1.** **According To Voltage Gaiting: Leishmania, Schistosome, Trypanosome** L-type high voltage calcium channel Interruption by Dihydropyridine Calcium Channel Blockade **2.** **According To Ligand: Leishmania, trypanosome** Sphingosine Inositol triphosphate with normal Ca^2+^ concentration Interruption by Miltefosine **3.** **According To Transient Receptor Potential: Toxoplasma, Schistosome** Normal intake/normal calcium level Interruption by Praziquantel/increased intake paralysis **4.** **According To Store-Operated Entry: Taenia** Interruption by increased calcium intake leading to apoptosis **5.** **According To Mono Or Divalent Ion Exchangers (Ca^2+^ dependent protein kinases)** Sodium/Calcium Plasmodium Sodium/Hydrogen/Calcium Toxoplasma Sodium/Acetylcholine Ascaris

**Table 6 biomedicines-14-00002-t006:** Parasites expressing calcium channels/ion exchangers.

A. Ascaris	Sodium/acetylcholine ion pump
B. Leishmania	L-type voltage-gated calcium channel
C. Plasmodium	Sodium/calcium ion exchanger
D. Schistosoma	Transient receptor potential channel
E. TaeniaW	Store-operated calcium entry channel
F. Toxoplasma	Transient receptor potential channel
	Sodium/hydrogen/calcium exchanger
G. Trypanosome	L-type voltage-gated calcium channel
	Ligand-related calcium channel
	Sphingosine
	Inositol triphosphate
	Store-operated calcium entry channels
	Acidocalcisomes (phosphate-containing organelles)
	Calcium-containing protein kinases
	Endoplasmic reticulum

## Data Availability

No new data were created or analyzed in this study. Data sharing is not applicable to this article.

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
