# Peer review of "Monocellular and Multicellular Parasites Infesting Humans: A Review of Calcium Ion Mechanisms"

_biomedicines, 2025, doi:10.3390/biomedicines14010002_

Round 1

Reviewer 1 Report

Comments and Suggestions for Authors

Dear Editor

The manuscript "Monocellular and Multicellular Pathogens Infesting Humans: A Review of Calcium Ion Channel Mechanisms" could have been more interesting, in case the authors design the article focused in single domain. There are several concerns regarding the concept- write-up and conclusion.

The title is misleading "Monocellular and Multicellular Pathogens Infesting Humans".... it could be Monocellular and Multicellular Parasites Infesting Humans.

In the current format of title: the abstract does not provide sufficient information.

Line 20-24: these lines does support title, hypotheses or provide any information related to title.

Similarly, the lines 25-33: these are not related to calcium channels or current title.

Line 34-35: "These therapies, identified for more than a 34 century (e.g. acidification of urine to potentiate treatment of urinary infections, use of 35 calcium channel blockade to increase circulating cyclosporine levels, neprilysin inhibition 36 to potentiate angiotensin receptor blockade or ice when added to nonsteroidals and rest)". The data is incorrect and without any citations.

Line 39: There is no reason to include Tuberculosis here.

Line 49-51: There is no reason to add avian malaria as it cited here in terms of mortality.

Lines 58-59: The concept of describing cystic fibrosis is not clear. Further there is redundancy in the whole paragraph.

Similarly, there are lot of further points, the authors did not concentrate on the topic and could not provide their clear and critical point of view.

Comments on the Quality of English Language

None

Author Response

The manuscript "Monocellular and Multicellular Pathogens Infesting Humans: A Review of Calcium Ion Channel Mechanisms" could have been more interesting, in case the authors design the article focused in single domain. There are several concerns regarding the concept- write-up and conclusion.

The title is misleading "Monocellular and Multicellular Pathogens Infesting Humans".... it could be Monocellular and Multicellular Parasites Infesting Humans.

Response: We found the critique helpful and changed the title and focus of the manuscript

In the current format of title: the abstract does not provide sufficient information.

Response: The abstract has been reimagined and information has been added

Line 20-24: these lines does support title, hypotheses or provide any information related to title.

Similarly, the lines 25-33: these are not related to calcium channels or current title.

Line 34-35: "These therapies, identified for more than a 34 century (e.g. acidification of urine to potentiate treatment of urinary infections, use of 35 calcium channel blockade to increase circulating cyclosporine levels, neprilysin inhibition 36 to potentiate angiotensin receptor blockade or ice when added to nonsteroidals and rest)". The data is incorrect and without any citations.

Response: The above lines have been eliminated

Line 39: There is no reason to include Tuberculosis here.

Response: Agree, deleted

Line 49-51: There is no reason to add avian malaria as it cited here in terms of mortality.

Lines 58-59: The concept of describing cystic fibrosis is not clear. Further there is redundancy in the whole paragraph.

Response: Agree, deleted

Similarly, there are lot of further points, the authors did not concentrate on the topic and could not provide their clear and critical point of view.

Response: Agree. We have rewritten major portions of this manuscript in keeping with the above comments

Reviewer 2 Report

Comments and Suggestions for Authors

In this manuscript, the authors review the importance of biological calcium concentration and calcium ion channels in the context of human–pathogen interactions, with a focus on parasites such as Plasmodium, Toxoplasma, Taenia, and Leishmania. The authors raise an important point regarding the potential of modulating calcium signaling as an adjuvant strategy to improve drug efficacy and reduce the emergence of resistance, which is indeed timely and of interest to both clinicians and clinical researchers. However, the current version of the manuscript falls short in several areas:

  1. The text reads as a compilation of information rather than a structured review. The manuscript would benefit from a clearer storyline that connects the biology of calcium signaling with the implications for parasite survival, host–pathogen interactions, and therapeutic potential.
  2. Several sentences are convoluted and imprecise, making the manuscript difficult to follow and potentially disengaging for the reader. Careful editing is required to improve clarity and flow.
  3. Line 110: The statement “… we might assume that…. infested” is misleading. In infections such as toxoplasmosis, it is well established (not assumed) that the parasite remains dormant unless the immune system is compromised. Such inaccuracies should be corrected to avoid confusing readers.
  4. Certain sections, for example Section 3, do not align with their subtitles. The subsections should provide a clear overview of the relevant host or parasite cellular pathways that are directly associated with intra- or extracellular calcium ions. Without this, the implications of these sections remain unclear.

To be suitable for publication, the manuscript must be reorganized to present the information in a cohesive and logical manner, while clearly highlighting the biological and therapeutic significance of calcium signaling in parasitic infections. In addition, a summary should be included to rationally justify the importance of calcium signaling–modulating agents as potential adjunct therapies.

Comments on the Quality of English Language

There are numerous sentences that are unclear and do not convey the intended message with precision. Several errors are evident in the use of scientific nomenclature, as well as in spelling and grammar, which need careful correction. In addition, the manuscript contains multiple instances where sentences are disconnected, leading to a lack of logical flow and coherence across sections/information.

Author Response

In this manuscript, the authors review the importance of biological calcium concentration and calcium ion channels in the context of human–pathogen interactions, with a focus on parasites such as Plasmodium, Toxoplasma, Taenia, and Leishmania. The authors raise an important point regarding the potential of modulating calcium signaling as an adjuvant strategy to improve drug efficacy and reduce the emergence of resistance, which is indeed timely and of interest to both clinicians and clinical researchers. However, the current version of the manuscript falls short in several areas:

  1. The text reads as a compilation of information rather than a structured review. The manuscript would benefit from a clearer storyline that connects the biology of calcium signaling with the implications for parasite survival, host–pathogen interactions, and therapeutic potential.
  2. Several sentences are convoluted and imprecise, making the manuscript difficult to follow and potentially disengaging for the reader. Careful editing is required to improve clarity and flow.
  3. Line 110: The statement “… we might assume that…. infested” is misleading. In infections such as toxoplasmosis, it is well established (not assumed) that the parasite remains dormant unless the immune system is compromised. Such inaccuracies should be corrected to avoid confusing readers.
  4. Certain sections, for example Section 3, do not align with their subtitles. The subsections should provide a clear overview of the relevant host or parasite cellular pathways that are directly associated with intra- or extracellular calcium ions. Without this, the implications of these sections remain unclear.

To be suitable for publication, the manuscript must be reorganized to present the information in a cohesive and logical manner, while clearly highlighting the biological and therapeutic significance of calcium signaling in parasitic infections. In addition, a summary should be included to rationally justify the importance of calcium signaling–modulating agents as potential adjunct therapies.

Comments on the Quality of English Language

There are numerous sentences that are unclear and do not convey the intended message with precision. Several errors are evident in the use of scientific nomenclature, as well as in spelling and grammar, which need careful correction. In addition, the manuscript contains multiple instances where sentences are disconnected, leading to a lack of logical flow and coherence across sections/information.

Response: In restructuring this manuscript we have eliminated our somewhat convoluted sentence structure and tried to write a more direct “storyline”. We hope that the clarity has been improved and our message has become clear. We have (we think) identified the calcium ion pathways for both the pathogen and human host without becoming too complex for the reader. Our hope is to encourage further research into an area that may be a target for further research and development for adjunctive therapies.

Reviewer 3 Report

Comments and Suggestions for Authors

The manuscript by D'Elia and Weinrauch presents a review of calcium channels and their role in pathogen-human interactions. The subject matter is relevant and the references appropriate.

The manuscript has a tendency however to wander between subjects and could use a bit more organization. Movement between muscle, blood, and different pathogens occurs within the manuscript without transition, and discussions on sickle cell anemia didn't seem very relevant.

A summary figure illustrating the working examples discussed in the text would also be helpful in visually framing the concepts.

A better description of the function of calcinerin would also be helpful. Section 6 could be more strongly tied to Ca2+.

Line 306 "would be useful as ammunition", - is this speculation or stated by the authors of the paper?

Comments on the Quality of English Language

There are a few grammar mistakes:

line 45

line 134

line 144

line 188

Author Response

The manuscript by D'Elia and Weinrauch presents a review of calcium channels and their role in pathogen-human interactions. The subject matter is relevant and the references appropriate.

The manuscript has a tendency however to wander between subjects and could use a bit more organization. Movement between muscle, blood, and different pathogens occurs within the manuscript without transition, and discussions on sickle cell anemia didn't seem very relevant.

Response: We accept this as a very valid criticism on reread of what we had submitted. As a result, we have rewritten and reorganized the manuscript in what we hope gives more clarity to our subject. Ee would hope that the reviewer will accept that the discussion of genetic susceptibility to the severe effects of infestation deserved some discussion. We hope that the clarity has been improved and our message has become clear. We have (we think) identified the calcium ion pathways for both the pathogen and human host without becoming too complex for the reader. Our hope is to encourage further research into an area that may be a target for further research and development for adjunctive therapies.

A summary figure illustrating the working examples discussed in the text would also be helpful in visually framing the concepts.

Response: We were unable to envisage a figure that would lend clarity and precision to all of the interactions discussed

A better description of the function of calcineurin would also be helpful. Section 6 could be more strongly tied to Ca2+.

Line 306 "would be useful as ammunition", - is this speculation or stated by the authors of the paper?

Response: This comment has been eliminated

Comments on the Quality of English Language

There are a few grammar mistakes:

Response these have been addressed

Reviewer 4 Report

Comments and Suggestions for Authors

The review article tries to offer information regarding calcium ion mechanisms and
his role on monocellular and multicellular parasites infesting humans. However, the information provided offers a summary of calcium movement in cell defense
systems, calcium movements outside and inside of the cell membrane, calcium
movement in cell immunity, the impact of calcium channel blockade and calcineurin inhibition on antibiotic drug resistance, calcium movements in parasite functions, calcium-related mechanisms in multicellular parasite infestations of vertebrate hosts and targeting calcium ion signalling and exchange.
The review lacks of clear conclusion stating the importance of the role of this
calcium ion channels with the parasites. Furthermore, they do not provide a clear
vision of how many calcium ion channels are or to which parasites are included in
this study. I do believe they have make a big effort but it can be improved to offer a better and more organised manuscript that fulfils the journal requeriments.

I consider the topic interesting and relevant. However, the manuscript could be
improved by making some amends such as: 1) definition and list of calcium ion
channels present in the parasites of interest, 2) a list of parasites covered in this
revision and 3) more clear presentation of their findings relating to the subject.
It provides a compilation useful (but could be improved) to work on, especially for
the development of chemical probes or drugs targeting calcium ion channels.

The review does not present all the available information in full or at least the
information provided does not stand that it comprises all of the data available.
However, is an interesting topic that should be developed.

Main comments:

The authors should firstly clarify in one paragraph the presence of calcium ion
channels on monocellular and multicellular parasites causing infections to humans
and their relevance or competition in biological process of the host. Thereafter, the different kinds of calcium ion channels and their roles can be better introduced and related to the parasites, not only by given examples but describing their possible competition and role inside the host. On the other hand, it would also be convenient to provide a short paragraph with the parasites to which the study is limited to. This will provide a clarification to the readers and make the manuscript more understandable.

The conclusions are consistent but limited to the channels presented in the
manuscript. As they do not provide an extensive list of the calcium ion channels that can be present, the reader is limited to what they present. The information could be improve by offering a more detailed revision of the calcium ion channels, and their relation with the parasites of interest mentioned.

Additional comments:

In the Abstract, there are some extra letters o words. Extra “of” in the text: Alteration of calcium ion channels responsive to environmental stimuli (Transient Receptor Potential) results in erythrocyte protection of from plasmodium. There is also one extra “s” in Echinococcus s,

Introduction line 5 correct the nomenclature of calcium ion to Ca2+and introduce the abbreviature, there is no reference in the text to indicate that Ca corresponds to calcium.

Parasites and mosquito scientific names are not in cursive in the first paraghraph (Number 1), whereas in number 2 they are redacted in cursive.

Anecdotal reports of the Anopheles mosquito in southern locations of the United States of America will eventually result in reappearance of clinical malaria after an infestation-free period of 70 years. This text needs to be referenced.

Nomenclature should also be uniformised, in some text calcium ion is referred as Ca2+ whereas in other (number 7) is presented as Ca++, use the same recommendation for the others ions.

Bibliography needs to be uniformized according to the guidelines of the journal, some of them shows the doi information while others don´t.

In table 1 the title should be amended affected instead of effected. Moreover, only two affected viscera are indicated (i.e., the affected by S. haematobium and Cryptosporidium), whereas the other examples have no precision on which viscera are affected. There is any reason to specify only two of them? It is suggested to uniformise the information presented.

In table 4, names of parasitic should be written in cursive.

Author Response

Main comments:

The authors should firstly clarify in one paragraph the presence of calcium ion channels on monocellular and multicellular parasites causing infections to humans and their relevance or competition in biological process of the host. Thereafter, the different kinds of calcium ion channels and their roles can be better introduced and related to the parasites, not only by given examples but describing their possible competition and role inside the host. On the other hand, it would also be convenient to provide a short paragraph with the parasites to which the study is limited to. This will provide a clarification to the readers and make the manuscript more understandable. 

Response: 

We have endeavored to more clearly separate the calcium ion channels in the parasite from those of the host on our presentation. Given the breadth of the subject, it is impossible within a single manuscript. Each of the parasites have different effects and systems that they attack. We have previously touched on the issue of host-parasite and host M tuberculosis interactions in Int J Mol Sci. 2024 Sep 10;25(18):9775 and Int J Mol Sci. 2023 Jun 2;24(11):9670. In an effort to make this manuscript more readable and understandable we have given it a considerable revision and added the sections that are highlighted with more than a dozen additional references (while eliminating some references that were no longer appropriate to the discussion with its changes).

Tables 4,5,6 attempt to present a summary of calcium -related functions in the human parasites selected. Table 4 is our perspective. Table 6 is our understanding of the direction preferred by reviewers. Table 5 is a hybrid of our point of view along with our understanding of the reviewer suggestions. WE jave responded to the reviewer’s suggestions by strengthening the conclusion.

The conclusions are consistent but limited to the channels presented in the manuscript. As they do not provide an extensive list of the calcium ion channels that can be present, the reader is limited to what they present. The information could be improve by offering a more detailed revision of the calcium ion channels, and their relation with the parasites of interest mentioned.                          

Response: We agree that the channels discussed are those for which there is some reliable literature and for which we have some knowledge of their interactions with respect to known and available medications. We hypothesize that currently available medications may have benefits in parasitic infestations. To go beyond that comment is perhaps a step too far without additional scientific information.

Additional comments:

In the Abstract, there are some extra letters o words. Extra “of” in the text: Alteration of calcium ion channels responsive to environmental stimuli (Transient Receptor Potential) results in erythrocyte protection of from plasmodium. There is also one extra “s” in Echinococcus s,                

Response: Corrected

Introduction line 5 correct the nomenclature of calcium ion to Ca2+and introduce the abbreviature, there is no reference in the text to indicate that Ca corresponds to calcium.                        

Response: Corrected

Parasites and mosquito scientific names are not in cursive in the first paraghraph (Number 1), whereas in number 2 they are redacted in cursive.                                                                          

Response: Corrected

Anecdotal reports of the Anopheles mosquito in southern locations of the United States of America will eventually result in reappearance of clinical malaria after an infestation-free period of 70 years. This text needs to be referenced.                                                                                                  

Response: Corrected

Nomenclature should also be uniformised, in some text calcium ion is referred as Ca2+ whereas in other (number 7) is presented as Ca++, use the same recommendation for the others ions.      

Response: Corrected

Bibliography needs to be uniformized according to the guidelines of the journal, some of them shows the doi information while others don´t.                                                                                  

Response: Corrected

In table 1 the title should be amended affected instead of effected. Moreover, only two affected viscera are indicated (i.e., the affected by S. haematobium and Cryptosporidium), whereas the other examples have no precision on which viscera are affected. There is any reason to specify only two of them? It is suggested to uniformise the information presented.

Response: Corrected

In table 4, names of parasitic should be written in cursive.

Response: Corrected

Round 2

Reviewer 1 Report

Comments and Suggestions for Authors

Dear Editor

The Authors have improved the article, unfortunately, the manuscript still does not contain sufficient novelty in its current form. A single mechanism could not be considered.

Author Response

We thank each of the reviewers for their constructive comments.

We do understand that the viewpoints of clinicians often are often expressed in different manners than those of some basic scientists. But, after carefully reviewing the grammar that we (native English speakers) composed we believe that the current manuscript does express our meanings. We did find an errant semi-colon or two. We do understand that this difficult topic is not reducible to a single brief hypothesis that will point all readers toward a specific target that will focus all researchers clearly. Our hopes are that our review will enable those who wish to study further to quickly find pertinent studies on which to base their scientific endeavors.

There are no other robust reviews of this topic appeqaring in widely published scientific literature. If we have neglected to reference any publications (save those we published or referenced over several decades regarding calcium ion metabolism, infections, sepsis, and immune suppressed populations), we would be happy to add such references to the current manuscript. As best stated by the journal’s reviewer #2 our goal is to “raise an important point regarding the potential of modulating calcium signaling as an adjuvant strategy to improve drug efficacy and reduce the emergence of resistance, which is indeed timely and of interest to both clinicians and clinical researchers.”

We hope that these comments are responsive to your requests and that the manuscript can now be published.

Reviewer 2 Report

Comments and Suggestions for Authors

The authors have appropriately addressed the previous concerns and the revised manuscript is well organized and improved. The current revised manuscript should be of interest to the readers interested in the topic.

Author Response

We thank each of the reviewers for their constructive comments.

We do understand that the viewpoints of clinicians often are often expressed in different manners than those of some basic scientists. But, after carefully reviewing the grammar that we (native English speakers) composed we believe that the current manuscript does express our meanings. We do understand that this difficult topic is not reducible to a single brief hypothesis that will point all readers toward a specific target that will focus all researchers clearly. Our hopes are that our review will enable those who wish to study further to quickly find pertinent studies on which to base their scientific endeavors.

There are no other robust reviews of this topic appeqaring in widely published scientific literature. If we have neglected to reference any publications (save those we published or referenced over several decades regarding calcium ion metabolism, infections, sepsis, and immune suppressed populations), we would be happy to add such references to the current manuscript. As best stated by the journal’s reviewer #2 our goal is to “raise an important point regarding the potential of modulating calcium signaling as an adjuvant strategy to improve drug efficacy and reduce the emergence of resistance, which is indeed timely and of interest to both clinicians and clinical researchers.”

We hope that these comments are responsive to your requests and that the manuscript can now be published.